# Consideration of Migraines Among Risk Factors for Postoperative Nausea and Vomiting

**DOI:** 10.3390/jcm9103154

**Published:** 2020-09-29

**Authors:** Jong-Ho Kim, Man-sup Lim, Sang-Hwa Lee, Young-Suk Kwon, Jae Jun Lee, Jong-Hee Sohn

**Affiliations:** 1Department of Anesthesiology and Pain Medicine, Chuncheon Sacred Heart Hospital, Hallym University college of Medicine, Chuncheon 24253, Korea; poik99@hallym.or.kr (J.-H.K.); gettys@hallym.or.kr (Y.-S.K.); iloveu59@hallym.or.kr (J.J.L.); 2Institute of New Frontier Research, College of Medicine, Hallym University, Chuncheon 24253, Korea; 3Department of Medical Education, College of Medicine, Hallym University, Chuncheon 24253, Korea; ellemes@hallym.ac.kr; 4Department of Neurology, Chuncheon Sacred Heart Hospital, Hallym University College of Medicine, Chuncheon 24253, Korea; neurolsh@hallym.or.kr

**Keywords:** postoperative nausea and vomiting, migraine, general anesthesia

## Abstract

The impact of migraine on postoperative nausea and vomiting (PONV) is controversial, and few studies have focused on their relationship. Thus, we investigated the impact of migraine, among other risk factors, on PONV in a large retrospective study. We analyzed 10 years of clinical data from the Smart Clinical Data Warehouse of Hallym University Medical Center. PONV was defined as nausea or vomiting within the first 24 h after surgery. Patients diagnosed by a neurologist and with a history of triptan use before surgery were enrolled into the migraine group. We enrolled 208,029 patients aged > 18 years who underwent general anesthesia (GA), among whom 19,786 developed PONV within 24 h after GA and 1982 had migraine. Before propensity score matching, the unadjusted and fully adjusted odds ratios (ORs) for PONV in subjects with versus without migraine were 1.52 (95% confidence interval (CI), 1.34–1.72; *p* < 0.001) and 1.37 (95% CI, 1.21–1.56; *p* < 0.001), respectively. The OR for PONV in patients with migraine was also high (OR, 1.37; 95% CI, 1.13–1.66; *p* = 0.001) after matching. Our findings suggest that migraine is a significant risk factor for PONV.

## 1. Introduction

Postoperative nausea and vomiting (PONV) are the most frequent side effects of anesthesia [1]. PONV has been reported to occur in 10–30% of all surgical patients, and the rate is as high as 80% in high-risk patients [2]. Although PONV is almost always self-limiting and nonfatal, it can cause dehydration, electrolyte imbalance, suture tension and dehiscence, venous hypertension and bleeding, esophageal rupture, and life-threatening airway compromise [3,4]. Therefore, prophylaxis and management of PONV are crucial for optimizing patient outcomes.

The first step for prophylaxis of PONV is identifying the risk factors and high-risk populations. Many anesthesia, surgery, and patient-related factors can increase PONV risk [5]. Of these risk factors, female gender, a history of PONV, being the first-degree relative of a PONV patient, being a nonsmoker, history of motion sickness, age < 50 years, lengthy surgery, and postoperative opioid use are well-established [5,6,7].

After surgery, postoperative gastrointestinal (GI) tract dysfunction is common, and is associated with increased patient suffering and cost of care [8]. The GI tract can become distended after anesthesia and surgery [9], and nausea or vomiting is likely to occur in situations where there is dysfunction of the lower esophageal sphincter, such as in gastroesophageal reflux disease (GERD) [10,11]. Migraine is accompanied by nausea and vomiting, which is therefore one of the diagnostic criteria. Migraine is also associated with GI disorders, such as gastroparesis, GERD, and irritable bowel syndrome [12,13,14], and is often accompanied by various upper GI symptoms [15,16]. However, the impact of migraine on PONV is controversial, and few studies have focused on their relationship. Previous studies have reported that a history of migraine tends to influence postoperative nausea only [17,18]. Other studies showed that PONV is more frequent in patients with a migraine history [19,20]. However, these reports were limited by small sample sizes and observational and/or questionnaire-based designs, which carry a risk of recall bias. Thus, we investigated the impact of migraine, among other factors, on PONV in a large retrospective study using propensity score matching.

## 2. Methods

### 2.1. Subjects

We retrospectively analyzed big clinical data, i.e., data from the Smart Clinical Data Warehouse (Smart CDW) of Hallym University Medical Center (HUMC). The Smart CDW, based on the QlikView Elite Solution (Qlik, Lund, Sweden), is used at the five hospitals of the HUMC. It offers electronic medical record text data analysis and an integrated analysis of fixed data. We collected the clinical data of patients aged ≥ 18 years who had undergone surgery with general anesthesia (GA) at one of the five HUMC hospitals between January 2010 and October 2019. We excluded patients who were re-operated on within 24 h, were unconscious after surgery, underwent ventilator therapy after surgery, experienced nausea or vomiting before surgery, or had missing data in their medical records. This study was approved by the Clinical Research Ethics Committee of Chuncheon Sacred Heart Hospital, Hallym University (IRB No. 2020-01-002)

### 2.2. Migraine, PONV, and Covariates

PONV was defined as nausea or vomiting within the first 24 h after surgery. Patients with migraine were eligible for inclusion if they met all of the following criteria: aged ≥ 18 years, diagnosis of migraine by a board-certificated neurologist, ≥2 consecutive visits to the neurology department, and a history of triptan use before surgery. Triptan drugs are used specifically to treat migraine; those used at HUMC include almotriptan, frovatriptan, naratriptan sumatriptan, and zolmitriptan. We analyzed the effect of migraine, along with other known and putative risk factors, on PONV. Covariates included age, sex, body mass index (BMI), anesthesia duration, American Society of Anesthesiologists physical status, use of N_2_O and inhalation anesthetics for maintenance of anesthesia, patient-controlled analgesia after surgery, history of diabetes, history of smoking, use of antiemetics, opioids, steroid, or antibiotics, use of a Levin tube during and after surgery, laparoscopic surgery, and transfusion during surgery

### 2.3. Statistical Analysis

Continuous data are presented as means and standard deviations, and categorical data as frequencies and percentages. A *t*-test was performed to compare the continuous data of patients with and without PONV. Categorical data were analyzed by the chi-square test. First, odds ratios (ORs) with 95% confidence interval (CIs) were calculated for the occurrence of PONV within 24 h after surgery, for each variable, by logistic regression. The OR is a measure of the association between an exposure and an outcome, and in this study represents the likelihood of PONV given a particular exposure compared to the likelihood in its absence. Fully adjusted ORs for PONV were then calculated for each variable, including migraine.

As patients could not be randomized based on the presence of migraine, confounding and selection biases were accounted for by using propensity scores. The rationale for using, and method for calculating, propensity scores for exposure variables have been described previously [21,22]. In this study, performed propensity score matching was conducted for normal patients and those with migraine. Python (version 3.7; Anaconda Inc., Austin, TX, USA) and Pymatch (version 0.3.4; https://github.com/benmiroglio/pymatch) were used for propensity score matching. The propensity scores ranged from 0.07–0.87. All matched cases had scores within 0.0001 of each other, and the matching ratio was 1:1 (1982 migraine patients and 1982 normal controls).

The ORs for PONV in migraine patients were compared to those of the normal controls. In the analysis, covariates and propensity scores were used to calculate adjusted ORs. All P-values were two-sided, and a *p*-value < 0.05 was considered significant. SPSS software (version 24.0; IBM Corp., Armonk, NY, USA) was used for the statistical analyses.

## 3. Results

### 3.1. Subject Characteristics

In total, 208,029 patients aged > 18 years underwent GA between January 2010 and October 2019 at HUMC. After excluding 20,323 patients, 187,706 were included in the study. The following patients were excluded: 1383 patients who were re-operated on within 24 h, 6886 unconscious patients, 7997 patients who underwent ventilator therapy after surgery, 2080 patients who had nausea or vomiting before surgery, and 1977 patients who had missing data in their medical records. Among the included patients, 19,786 developed PONV within 24 h after GA and 1982 had migraine. Table 1 summarizes the baseline characteristics of the patients who received GA during surgery. Significant differences were observed in all clinical variables between the PONV and non-PONV patients, except age. PONV occurred in 300 of the 1982 (1.5%) patients with migraine (Table 1).

### 3.2. Odds Ratio for PONV in Migraineurs

Figure 1 shows the unadjusted and fully adjusted ORs for PONV of each clinical variable. The unadjusted OR for the occurrence of PONV in subjects with versus without migraine was 1.52 (95% CI, 1.34–1.72; *p* < 0.001). Fully adjusted ORs for the occurrence of PONV and migraine were obtained through a multivariate logistic analysis including all clinical variables and covariates. The fully adjusted OR for the occurrence of PONV in subjects with versus without migraine was 1.37 (95% CI, 1.21–1.56; *p* < 0.001).

No significant difference was observed in any clinical variable between the propensity score-matched migraine and normal patients (Table 2). The OR for PONV of migraine was high after propensity score matching (OR, 1.37; 95% CI, 1.13–1.66; *p* = 0.001) (Table 3).

## 4. Discussion

This study examined the impact of migraine on PONV in patients undergoing GA. We enrolled patients who had undergone GA at the HUMC over a 10-year period using the Smart CDW. In total, 208,029 patients aged > 18 years underwent GA, 19,786 developed PONV within 24 h after GA, and 1982 had migraine. The unadjusted and fully adjusted ORs for the occurrence of PONV in migraineurs were 1.52 (95% CI, 1.34–1.72; *p* < 0.001) and 1.37 (95% CI, 1.21–1.56; *p* < 0.001) relative to those without migraine, respectively. After propensity score matching, the OR for PONV of migraine remained high (OR, 1.37; 95% CI, 1.13–1.66; *p* = 0.001).

It is believed that PONV has a multifactorial origin and there are numerous risk factors. Risk factors may be classified as “well-established”, “possible”, or “disproved”, according to expert consensus and the opinion of the author, as well as in terms of whether they are patient-, surgery-, or anesthesia-related [5]. Established patient-related risk factors include female gender, nonsmoking status, younger age, and history of motion sickness or PONV. There are also various anesthesia- and surgery-related factors, such as general versus regional anesthesia, duration of anesthesia, use of volatile anesthetics, nitric oxide, and opioids, and type and duration of surgery [5,6,23]. However, few studies have assessed the effect of migraine on PONV, so their relationship is poorly understood. A previous questionnaire-based study on PONV, including 253 patients under GA, reported that PONV was more prevalent in patients with a migraine history [19]. In the only randomized-controlled study to assess the role of migraine history in the development of PONV, 127 women were randomized into groups according to the anesthesia method. Migraine history was a risk factor of PONV, particularly in the GA group, and epidural anesthesia did not affect the likelihood of PONV in women with migraine [20]. Other studies have reported that a history of migraine is more strongly associated with nausea than vomiting [17,18]. Prospective interview-based surveys on the incidence of PONV in surgical in-patients (1107 and 671 patients, respectively) under GA or local anesthesia have been conducted [24,25]. Among the patients in these two studies, 822 (74%) and 480 (72%) received GA, and 166 (15%) and 63 (9.3%) had a migraine history, respectively. Logistic regression and bivariate Dale models of PONV showed that a history of migraine was significantly related to nausea but not vomiting, which was explained by differences in physiological mechanisms [24,25]. Another study proposed that nausea and vomiting should be analyzed separately and considered as “two biologically different phenomena” [26]. Nausea is a subjective sensation involving the activation of certain neural pathways, which ultimately project to the areas of the cerebral hemispheres responsible for conscious sensation. Vomiting is a complex reflex under the control of two functionally distinct medullar centers: the vomiting center in the dorsal part of the lateral reticular formation, and the chemoreceptor trigger zone in the area postrema (in the floor of the fourth ventricle) [18]. However, our study did not consider nausea and vomiting as separate outcomes because they have identical risk factors and predictors [2,17,27,28,29,30,31]. Postoperative nausea is considered as a risk factor for vomiting [5], and there is a strong association between the two symptoms [18]. The neuroanatomical site controlling nausea and vomiting is an ill-defined region called the “vomiting center” within the reticular formation in the brainstem [32]. It receives afferent inputs from the chemoreceptor trigger zone, the vagal mucosal pathway in the GI system, neuronal pathways arising from the vestibular system, reflex afferent pathways arising from the cerebral cortex, and midbrain afferents; it also interacts with the nucleus tractus solitarius [33].

In most previous studies, patients were either asked about migraine symptoms directly or completed a self-reported questionnaire [17,18,19]. Few studies have used chart review for diagnosis. In our study, patients were diagnosed with migraine by board-certificated neurologists or based on ≥2 consecutive visits to the neurology department or a history of triptan drug use. Our study also included a large sample (>200,000 patients), which may have been more representative of patients undergoing GA referred for assessment of PONV than those of several small observational studies and randomized controlled trials that investigated the association between PONV and migraine [17,18,19,20].

PONV is typically associated with anesthesia or secondary to postoperative ileus, which is characterized by inhibition of GI motility following surgery [34]. Anesthesia causes GI distension [9]. All anesthetic and analgesic drugs also have the potential to contribute to postoperative GI tract dysfunction, including nitrous oxide, which accumulates in the body cavity and promotes bowel distension. Opioids have direct and indirect effects on bowel function, leading to decreased motility [35]. In turn, this can result in nausea and vomiting by increasing abdominal pressure and reflux. Anesthetics cause a highly significant pressure drop at the lower esophageal sphincter, and an adaptive change in the tone of the lower esophageal sphincter may occur with a general increase in abdominal pressure [36,37]. However, the adaptive change in the lower esophageal sphincter in response to increased intraabdominal pressure is abnormally small in patients with symptoms of reflux, such as GERD [38]. Previous studies have reported that GERD is associated with, and can predict, PONV [10,39].

Migraines are associated with various GI disorders, such as gastroparesis, GERD, irritable bowel syndrome, and celiac disease [12,13,14], and also cooccur with GI symptoms such as diarrhea, constipation, and dyspepsia [15,16]. A previous web-based survey, and a large CDW study of migraine patients, reported GERD prevalence rates of 22% and 27%, respectively [14,40]. Another clinic-based study reported that the prevalence of GERD was higher in migraineurs than non-migraineurs (42% vs. 18%) [41]. Migraine is often complicated by GI conditions, such as gastric stasis (also called gastroparesis), which is characterized by delayed emptying of the stomach in the absence of mechanical obstruction; its clinical manifestations include nausea, vomiting, and weight loss [42,43]. Previous studies reported gastric stasis in the interictal and ictal periods during spontaneous and induced migraine in migraineurs [44,45]. Furthermore, it was demonstrated empirically that migraineurs experience significant delays in gastric emptying, both during and outside of attacks, compared to non-migraine controls [12]. Taken together, these findings suggest that abnormal gastric motility is not just an acute event; instead, it is a clinical feature of migraine. Therefore, the GI disorders accompanying migraine may increase the incidence of PONV.

Calcitonin gene-related peptide (CGRP) also plays a central role in migraine and is the target of new migraine treatments. CGRP is a neuropeptide existing in two isoforms. The α isoform is implied in the pathogenesis of migraine pain, while the β isoform is primarily expressed by the enteric sensory system and contributes to the regulation of its motility [46,47]. Previous studies showed that CGRP per se may reduce gastric emptying [48,49]. In migraineurs, novel treatment with the monoclonal antibodies targeting the CGRP pathway can reduce GI motility and cause constipation [50,51]. Therefore, we suggest that the possible onset of PONV after surgery/anesthesia in patients with migraine is associated with relationship between CGRP and GI motility.

GI tract motor and secretory activities are controlled by a range of neural and hormonal systems. Neural control can be excitatory or inhibitory, local or central, and parasympathetic (predominantly excitatory) or sympathetic (mainly inhibitory) [8]. Autonomic nervous system dysfunction has been linked to both migraine and GI disorders [52,53,54,55,56,57], including in terms of pathogenesis; this may explain the overlap between migraine and GI disorders, such as GERD and gastric stasis. The pathophysiology of PONV is not fully understood. Nausea and vomiting may be induced through a variety of central and peripheral mechanisms. The cortex and vomiting center in the brainstem are influenced by various extrinsic factors, including movement (via the vestibular system), drugs (such as opioids and anesthetics), activity in the area postrema, intestinal vagal afferent fiber activity and pain (in association with surgical manipulation of the gut), and emotions during the perioperative period (mediated by higher cortical centers) [34]. Migraine is associated with a wide range of psychiatric comorbidities, as well as various vestibular symptoms and vestibular syndromes [58,59,60,61]. The pain experienced by migraine sufferers, and the accompanying symptoms such as dizziness, vertigo, and anxiety, are likely to affect the development of PONV.

Our study had several limitations. First, it used data from subjects who visited one university medical center with five hospitals. Therefore, it is difficult to generalize the results to the general population, and the possibility of selection bias must be considered. Second, our study used a retrospective design and we did not collect clinical data on headache characteristics, such as frequency and duration, or medication history (i.e., use of NSAIDs or analgesics). We classified as “migraineurs” if patients met all of the following criteria: diagnosis by a board-certificated neurologist, 2 ≥ visits to the neurology department, and a history of triptan use before surgery to select the accurate diagnosis of migraine. However, due to these strict classification criteria in the diagnosis of migraine, the possibility of selection bias, in which some migraine patients are even classified as controls, cannot be excluded. Finally, we did not analyze the surgeries separately except for laparoscopic surgery. Additional prospective population-based studies are needed to conform the association between migraine and PONV.

## 5. Conclusions

This study collected the clinical data of a large number of patients from the Smart CDW, in addition to data on numerous risk factors for PONV. The impact of migraine on PONV is controversial, but previous studies had the limitations of small samples and an observational and/or questionnaire-based design. Our findings suggest that migraine has a significant impact on the development of PONV in patients undergoing GA. However, more research is needed to confirm the impact of migraine on PONV.

## Figures and Tables

**Figure 1 jcm-09-03154-f001:**
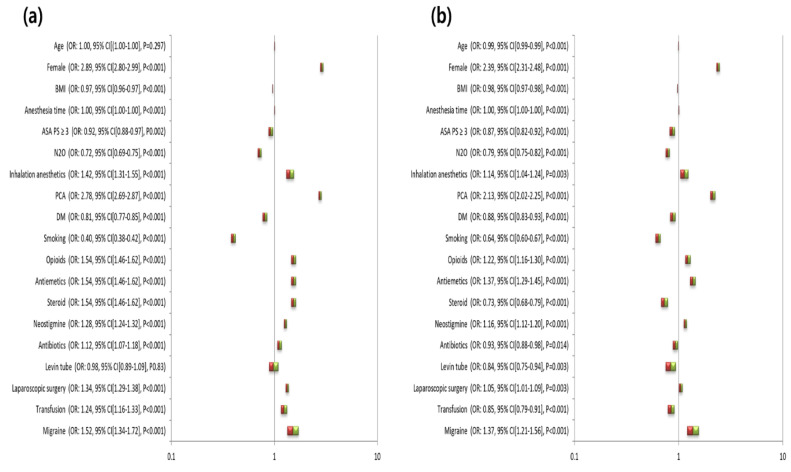
Unadjusted (**a**) and fully adjusted (**b**) odds ratios for postoperative nausea and vomiting of each variable. PONV, postoperative nausea and vomiting; BMI, body mass index; ASA PS, American Society of Anesthesiologists physical status; PCA, patient-controlled analgesia; DM, diabetes.

**Table 1 jcm-09-03154-t001:** Baseline characteristics of subjects.

	Non-PONV *n* = 167,920	PONV *n* = 19,786	*p*-Value
Age (Years, mean ± SD)	49.3 ± 16.9	49.5 ± 16.2	0.282
Female (*n* (%))	83,803(49.9)	14,696 (74.3)	<0.0001
BMI (Mean ± SD)	24.2 ± 3.8	23.8 ± 3.7	<0.0001
Anesthesia Time (Minutes, Mean ± SD)	135.7 ± 91.2	153.4 ± 89.8	<0.0001
ASA PS ≥ 3 (*n*, (%))	20,199 (12.0)	2227 (11.3)	0.002
N_2_O (*n* (%))	38,080 (22.7)	3473 (17.6)	<0.0001
Inhalation Anesthetics (*n* (%))	160,757 (95.7)	19,187 (97.0)	<0.0001
PCA (*n* (%))	73,583 (43.8)	13,548 (68.5)	<0.0001
DM (*n* (%))	19,589 (11.7)	1920 (9.7)	<0.0001
Smoking (*n* (%))	32,408 (19.3)	1731 (8.7)	<0.0001
Opioids (*n* (%))	146,743 (87.4)	18,091 (91.4)	<0.0001
Antiemetics (*n* (%))	90,457 (53.9)	14,573 (73.7)	<0.0001
Steroid (*n* (%))	10,828 (6.4)	796 (4.0)	<0.0001
Neostigmine (*n* (%))	47,437 (28.2)	6551 (33.6)	<0.0001
Antibiotics (*n* (%))	151,231 (90.1)	18,023 (91.1)	<0.0001
Levin tube (*n* (%))	3416 (2.0)	398 (2.0)	0.028
Laparoscopic surgery (*n* (%))	36,211 (21.6)	5333 (27.0)	<0.0001
Transfusion (*n*, (%))	6914 (4.1)	1005 (5.1)	<0.0001
Migraine (*n* (%))	1682 (1.0)	300 (1.5)	<0.0001

PONV, postoperative nausea and vomiting; SD, standard deviation; BMI, body mass index; ASA PS, American Society of Anesthesiologists physical status; PCA, patient-controlled analgesia; DM, diabetes.

**Table 2 jcm-09-03154-t002:** Characteristics of the two groups after propensity score matching.

	Migraine *n* = 1982	Normal *n* = 1982	D	*p*-Value
Age (years, mean ± SD)	47.82 ± 15.21	47.37 ± 16.80	0.03	0.373
Female (*n* (%))	1459 (73.6%)	1454 (73.4%)	0.01	0.857
BMI (mean ± SD)	24.57±4.39	24.48±4.11	0.01	0.510
Anesthesia time (Minutes, Mean ± SD)	121.30 ± 78.60	119.12 ± 73.95	0.03	0.369
ASA PS ≥ 3 (*n* (%))	309 (15.6%)	299 (15.1%)	0.02	0.659
N_2_O (*n*, (%))	321 (16.2%)	333 (16.8%)	0.02	0.608
Inhalation Anesthetics (*n* (%))	1892 (95.5%)	1901 (95.9%)	0.06	0.482
PCA (*n* (%))	778 (39.3%)	775 (39.1%)	<0.01	0.922
DM (*n* (%))	170 (8.6%)	159 (8.0%)	0.04	0.527
Smoking (*n* (%))	258 (13.0%)	292 (14.7%)	0.08	0.118
Opioids (*n* (%))	1787 (90.2%)	1795 (1795%)	0.03	0.667
Antiemetics (*n* (%))	1000 (50.5%)	1007 (50.8%)	0.01	0.824
Steroid (*n* (%))	134 (6.8%)	152 (7.7)	0.07	0.269
Neostigmine (*n* (%))	691 (34.9%)	663 (33.5%)	0.03	0.348
Antibiotics (*n* (%))	1776 (89.6%)	1778 (89.7%)	0.01	0.917
Levin tube (*n* (%))	16 (0.8%)	13 (0.7%)	0.10	0.576
Laparoscopic surgery (*n* (%))	405 (20.4%)	385 (19.4%)	0.03	0.426
Transfusion (*n* (%))	50 (2.5%)	57 (2.9%)	0.07	0.493

PONV, postoperative nausea and vomiting; BMI, body mass index; ASA PS, American Society of Anesthesiologists physical status; PCA, patient-controlled analgesia; DM, diabetes.

**Table 3 jcm-09-03154-t003:** Adjusted odds ratios for PONV of migraine after propensity score matching.

	Odds Ratio (95% CI)	*p*-Value
Unadjusted	1.36 (1.13–1.63)	0.001
Fully Adjusted	1.37 (1.13–1.66)	0.001
All Covariates + Propensity Score-Adjusted	1.37 (1.13–1.66)	0.001

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
