# Peer review of "Consideration of Migraines Among Risk Factors for Postoperative Nausea and Vomiting"

_jcm, 2020, doi:10.3390/jcm9103154_

Round 1

Reviewer 1 Report

In their case-control study, Authors assessed the association between migraine and postoperative nausea or vomiting. In multivariate analyses and after propensity score matching, migraine was indipendente associated with postoperative nausea or vomiting.

The paper is well written and potentially interesting. I only have some minor concerns.

  • How was the choice of covariates performed? Did Authors consider covariates previously assessed by the available studies?
  • The diagnosis of migraine in the present study might be a matter of concern. Authors state in the Methods that patients were classifica as migraineurs if meeting three criteria (triptan use, neurology visits, previous diagnosis); however, not all migraineurs have received a diagnosis by a neurologist, not all of them use triptans, and not all perform neurology visits. Therefore, some migraineurs might have been mister and even included as controls. This is a limitation to discuss.
  • Calcitonin gene-related peptide (CGRP) is a key mediator of migraine pain which is involved in the pathophysiology of migraine and also regulates gastrointestinal motility. This relationship can favor the onset of gastrointestinal adverse events due to the novel anti-migraine medications (please refer to J Headache Pain 2020;21:26 and Intern Emerg Med 2020 Jun 17, doi: 10.1007/s11739-020-02407-y). This point, in my opinion, deserves discussion.

Author Response

Dear Reviewer 1,

Please find attached a revised version of our manuscript, “Consideration of migraines among risk factors for postoperative nausea and vomiting” (jcm-907096).

We thank you for your thoughtful suggestions regarding the original version of our paper; most of the suggested changes have been incorporated into the revision.

All revisions are described in detail in the order mentioned in the review, following the reviewer’s critique in italics. We believe that the revisions have greatly improved the manuscript and hereby submit the revised version for your consideration for publication.

Comments to author:

In their case-control study, Authors assessed the association between migraine and postoperative nausea or vomiting. In multivariate analyses and after propensity score matching, migraine was indipendente associated with postoperative nausea or vomiting.

The paper is well written and potentially interesting. I only have some minor concerns.

We thank the reviewer for these comments and specific suggestions, which have improved our manuscript.

How was the choice of covariates performed? Did Authors consider covariates previously assessed by the available studies?

Thank you for your comment. Many studies have documented the risk factors for PONV among patients who have undergone anesthesia and surgery. We analyzed known and expected risk factors for PONV by previous studies, including age, sex, body mass index (BMI), anesthesia duration, American Society of Anesthesiologists physical status, use of N2O and inhalation anesthetics for maintenance of anesthesia, patient-controlled analgesia after surgery, history of diabetes, history of smoking, use of antiemetics, opioids, steroid, or antibiotics, use of a Levin tube during and after surgery, laparoscopic surgery, and transfusion during surgery. Also, we analyzed the effect of gastroesophageal reflex disease or BMI on PONV in our previous studies (J Clinical Med 2020, 9, 825; J Clinical Med 2020,9,1612). In each analysis, we used similar covariates to calculate the adjusted odds ratio.

The diagnosis of migraine in the present study might be a matter of concern. Authors state in the Methods that patients were classifica as migraineurs if meeting three criteria (triptan use, neurology visits, previous diagnosis); however, not all migraineurs have received a diagnosis by a neurologist, not all of them use triptans, and not all perform neurology visits. Therefore, some migraineurs might have been mister and even included as controls. This is a limitation to discuss.

We agree with this important comment. The study was retrospective in nature and based on medical records from our clinical data warehouse. We classified as migraineurs if they met all of the following criteria: diagnosis of migraine by a bord-certificated neurologist, 2 ≥ visits to the neurology department, and a history of triptan use before surgery in this study to select the accurate diagnosis of migraine in this study. However, due to these strict classification criteria in diagnosis of migraine, the possibility of selection bias, in which some migraine patients are even classified as controls, cannot be excluded. We now include a limitations in the Discussion. The text has been revised as follows.

Also, we classified as migraineurs if they met all of the following criteria: diagnosis by a bord-certificated neurologist, 2 ≥ visits to the neurology department, and a history of triptan use before surgery to select the accurate diagnosis of migraine. However, due to these strict classification criteria in diagnosis of migraine, the possibility of selection bias, in which some migraine patients are even classified as controls, cannot be excluded. (page 7, lines 240–lines 245)

Calcitonin gene-related peptide (CGRP) is a key mediator of migraine pain which is involved in the pathophysiology of migraine and also regulates gastrointestinal motility. This relationship can favor the onset of gastrointestinal adverse events due to the novel anti-migraine medications (please refer to J Headache Pain 2020;21:26 and Intern Emerg Med 2020 Jun 17, doi: 10.1007/s11739-020-02407-y). This point, in my opinion, deserves discussion.

We appreciate for the reviewer’s comments. We have added the following sentences in the Discussion and have added 6 articles to the references as follows.

Also, calcitonin gene-related peptide (CGRP) plays a central role in migraine and is the target of new migraine treatments. CGRP is a neuropeptide existing in two isoforms. The α isoform is implied in the pathogenesis of migraine pain, while the β isoform is primarily expressed by the enteric sensory system and contributes to the regulation of its motility [46,47]. Previous studies showed that CGRP per se may reduce gastric emptying [48,49]. In migraineurs, novel treatment with the monoclonal antibodies targeting the CGRP pathway can reduce GI motility and cause constipation [50,51]. Therefore, we suggest that the possible onset of PONV after surgery/anesthesia in patients with migraine is associated with relationship between CGRP and GI motility. (page 6, lines 214–lines 221)

  1. Deen, M; Correnti, E; Kamin, K; Keiderman, T; Papeti, L; Rubio Beltran, E; Vigneri, S; Edvisson, L; Maassen Van Den Brink A. On the behalf of the European Headache Federation School of Advanced Studies (EHF-SAS) Blocking CGRP in migraine patients-a review of pros and cons. J Headache Pain 2017, 18, 96.
  2. Tiseo, C; Ornello, R; Pistoia, F; Sacco,S. How to integrate monoclonal antibodies targeting the calcitonin gene-related peptide or its receptor in daily clinical practice. J Headache Pain 2019, 20, 49.
  3. Jurgen, LH. Calcitonin and CGRP inhibit gastrointestinal transit via distinct neuronal pathways. Am J Phys 1988, 254, G920-G924.
  4. L’Heureux, MC; St-Pierre, S; Trudel, L; Plourde, V; Lepage, R; Poitraus P. Digestive motor effects and vascular actions of CGRP in dog are expressed by different receptor subtypes. Peptides 2000, 21, 425-430.
  5. Frattale, L; Ornello, R; Pistoia, F; Caponnetto, V; Colangeli, E; Sacco, S. Paralytic ileus after planned abdominal surgery in a patient on treatment with erenmab. Intern Emerg Med 2020, 17, doi: 10.1007/s11739-020-02407-y.
  6. Haanes, KA; Edvinsson, L; Sams, A. Understanding side-effects of anti-CGRP and anti-CGRP receptor antibodies. J Headache Pain 2020, 21, 26.

We have tried to address the issues raised by the reviewers and editorial board member. We are grateful for the constructive comments that arose during the review process. We believe that our paper has been improved based on these suggestions.

Yours faithfully,

Reviewer 2 Report

The authors present large retrospective study on postoperative nausea and vomiting (PONV) after general anesthesia among 19.786 patients from the group of 208.029 patients, from whom 1982 had migraine. Basing on received findings and odds ratios for PONV in patients with migraine they concluded that migraine was a significant risk factor for PONV

The authors described the important role of anesthesia and surgery-related factors. 

For instance, what was the impact of the type of surgery on PONV?  

Could it be omitted as a meaningful factor e.g.: ear surgery, laryngeal surgery, posterior fossa neurosurgical intervention, abdominal surgery, cholecystectomy, laparoscopic surgery (the only variable) ?

In these cases, afferents of vomiting center in reticular formation of brainstem can be activated.

Is PONV related to the pathophysiology of migraine or is more associated with triptans use, in the authors’ opinion? Was the measured OR for PONV associated with migraine or exposure to triptans ?

This study collects the data on numerous risk factors of PONV and migraine was selected due to small amount of previous reports and low evidence of migraine as a risk factor of PONV.

Odds ratio for migraine is similar as for other factors as antiemetics, opioids, inhalation anesthetics. Migraine has been highlighted because its role has not been examined yet.

            There are clear disproportions in the numbers of patients with different predisposing factors to PONV. For instance, 300 patients with migraine had PONV, in comparison to 1682 with migraine who had no PONV, meanwhile PONV occurred in 19.187 patients with inhalation anesthetics, 18.091 patients with opioids, 18.023 with antibiotics,  14.573 patients, who received antiemetics, which seems obvious. 

In the context of dominant impact of other risk factors for PONV,  I am not sure if migraine deserves for such a distinction. From 208.029 patients after surgery 300 patients had migraine and PONV. PONV was present in 19.786 patients. The authors conclude that these findings suggest that migraine is a significant risk factor for PONV.

The title describing in general risk factors of postoperative nausea and vomiting would be more relevant in my opinion.

Author Response

Dear Reviewer 2

Please find attached a revised version of our manuscript, “Consideration of migraines among risk factors for postoperative nausea and vomiting” (jcm-907096).

We thank you for your thoughtful suggestions regarding the original version of our paper; most of the suggested changes have been incorporated into the revision.

All revisions are described in detail in the order mentioned in the review, following the reviewer’s critique in italics. We believe that the revisions have greatly improved the manuscript and hereby submit the revised version for your consideration for publication.

Comments to author:

The authors present large retrospective study on postoperative nausea and vomiting (PONV) after general anesthesia among 19.786 patients from the group of 208.029 patients, from whom 1982 had migraine. Basing on received findings and odds ratios for PONV in patients with migraine they concluded that migraine was a significant risk factor for PONV

We thank the reviewer for these comments and specific suggestions, which have improved our manuscript.

The authors described the important role of anesthesia and surgery-related factors. For instance, what was the impact of the type of surgery on PONV? Could it be omitted as a meaningful factor e.g.: ear surgery, laryngeal surgery, posterior fossa neurosurgical intervention, abdominal surgery, cholecystectomy, laparoscopic surgery (the only variable)?

In these cases, afferents of vomiting center in reticular formation of brainstem can be activated.

Thank you for your comment. Studies of the effect of type of surgery on the incidence of PONV have yielded conflicting results. Strabismus surgery, adenotonsillectomy, and inguinal scrotal and penile procedures have been reported as risk factors for PONV. However, we did not analyze the surgeries separately because these previous studies were mostly performed in pediatric patients and thus the results cannot be directly compared to our adult cohort. A systemic review of PONV suggested that laparoscopic surgery is associated with increased risk of PONV. Since most of the cholecystectomy and gynecological surgery are performed by laparoscopy, laparoscopic surgery, a well-known risk factor, was included as the variable in this study. Although the effect of the type of surgery on the prevalence of PONV is still discussed as one of the controversial issues due to the effect of the patients' characteristics and the factors associated with anesthesia, many large trials showed no relationship between the type of surgery and prevalence of PONV. In this study, we did not analyze the surgeries according to the type of operation, except for laparoscopic surgery. Thus, we have added the following text to the limitations in the Discussion section as follows

Finally, we did not analyze the surgeries separately except for laparoscopic surgery. (page 7, lines 245)

Is PONV related to the pathophysiology of migraine or is more associated with triptans use, in the authors’ opinion? Was the measured OR for PONV associated with migraine or exposure to triptans?

We thank you for the comments. The impact of migraine on PONV is controversial, but previous studies had the limitations of small samples and an observational and/or questionnaire-based design. The OR for PONV of migraine was high after propensity score matching. Our findings suggest that migraine has a significant impact on the development of PONV in patients undergoing GA. We suggest that PONV is related to the pathophysiology of migraine. Calcitonin gene-related peptide (CGRP) is a key mediator of migraine pain which is involved in the pathophysiology of migraine and also regulates gastrointestinal motility. We suggest that the possible onset of PONV after surgery/anesthesia in patients with migraine is associated with relationship between CGRP and GI motility. Thus, we have added the following sentences in the Discussion section and have added 6 articles to the references a follows.

Also, calcitonin gene-related peptide (CGRP) plays a central role in migraine and is the target of new migraine treatments. CGRP is a neuropeptide existing in two isoforms. The α isoform is implied in the pathogenesis of migraine pain, while the β isoform is primarily expressed by the enteric sensory system and contributes to the regulation of its motility [46,47]. Previous studies showed that CGRP per se may reduce gastric emptying [48,49]. In migraineurs, novel treatment with the monoclonal antibodies targeting the CGRP pathway can reduce GI motility and cause constipation [50,51]. Therefore, we suggest that the possible onset of PONV after surgery/anesthesia in patients with migraine is associated with relationship between CGRP and GI motility. (page 6, lines 214–lines 221)

  1. Deen, M; Correnti, E; Kamin, K; Keiderman, T; Papeti, L; Rubio Beltran, E; Vigneri, S; Edvisson, L; Maassen Van Den Brink A. On the behalf of the European Headache Federation School of Advanced Studies (EHF-SAS) Blocking CGRP in migraine patients-a review of pros and cons. J Headache Pain 2017, 18, 96.
  2. Tiseo, C; Ornello, R; Pistoia, F; Sacco,S. How to integrate monoclonal antibodies targeting the calcitonin gene-related peptide or its receptor in daily clinical practice. J Headache Pain 2019, 20, 49.
  3. Jurgen, LH. Calcitonin and CGRP inhibit gastrointestinal transit via distinct neuronal pathways. Am J Phys 1988, 254, G920-G924.
  4. L’Heureux, MC; St-Pierre, S; Trudel, L; Plourde, V; Lepage, R; Poitraus P. Digestive motor effects and vascular actions of CGRP in dog are expressed by different receptor subtypes. Peptides 2000, 21, 425-430.
  5. Frattale, L; Ornello, R; Pistoia, F; Caponnetto, V; Colangeli, E; Sacco, S. Paralytic ileus after planned abdominal surgery in a patient on treatment with erenmab. Intern Emerg Med 2020, 17, doi: 10.1007/s11739-020-02407-y.
  6. Haanes, KA; Edvinsson, L; Sams, A. Understanding side-effects of anti-CGRP and anti-CGRP receptor antibodies. J Headache Pain 2020, 21, 26.

This study collects the data on numerous risk factors of PONV and migraine was selected due to small amount of previous reports and low evidence of migraine as a risk factor of PONV.

Odds ratio for migraine is similar as for other factors as antiemetics, opioids, inhalation anesthetics. Migraine has been highlighted because its role has not been examined yet.

We thank you for the comments. The impact of migraine on PONV is controversial, but previous studies had the limitations of small samples and an observational and/or questionnaire-based design. Also, our study included a large sample (> 200,000 patients), which may have been more representative of patients undergoing GA referred for assessment of PONV than those of several small observational studies and randomized controlled trials that investigated the association between PONV and migraine. Our findings suggest that migraine has a significant impact on the development of PONV in patients undergoing GA. However, more research is needed to confirm the impact of migraine on PONV.

There are clear disproportions in the numbers of patients with different predisposing factors to PONV. For instance, 300 patients with migraine had PONV, in comparison to 1682 with migraine who had no PONV, meanwhile PONV occurred in 19.187 patients with inhalation anesthetics, 18.091 patients with opioids, 18.023 with antibiotics, 14.573 patients, who received antiemetics, which seems obvious. In the context of dominant impact of other risk factors for PONV, I am not sure if migraine deserves for such a distinction. From 208.029 patients after surgery 300 patients had migraine and PONV. PONV was present in 19.786 patients. The authors conclude that these findings suggest that migraine is a significant risk factor for PONV.

Thank you for your important comment. Of the risk factors to PONV, inhalation anesthetics, opioids, antibiotics and anti-emetic drugs are almost routinely used during anesthesia. There are fewer migraine patients, but the odds of the occurrence of PONV in migraine patients are high. In these disproportions in the numbers of patients with different predisposing factors to PONV, we try to evaluate the impact of migraine on PONV. Therefore, we performed propensity score matching (PSM) to reduce the bias of other factors. In statistical analysis of observational data, PSM is a statistical matching technique that attempts to estimate the effect of treatment, policy, or other intervention by considering the covariate predicted to receive treatment. We tried to reduce the bias due to confounding variables that can be found in estimates of treatment effectiveness obtained by simply comparing. For each covariate, randomization means that treatment groups are averagely balanced according to the law of large numbers. Unfortunately, for observational studies, treatment assignments to study subjects are generally not random. PSM is an attempt to reduce treatment allocation bias and mimic randomization by creating a treated unit sample that can be compared to an untreated unit sample at all observed covariates.

The title describing in general risk factors of postoperative nausea and vomiting would be more relevant in my opinion.

We appreciate for the reviewer’s correction. We changed the title as follows.

Consideration of migraines among risk factors for postoperative nausea and vomiting (page 1, lines 2-lines 3)

We have tried to address the issues raised by the reviewers and editorial board member. We are grateful for the constructive comments that arose during the review process. We believe that our paper has been improved based on these suggestions.

Yours faithfully,